# OpenReview forum: "Hybrid Quantum-Classical Policy Gradients for Multi-Agent Reinforcement Learning: A Principled Analysis of Expressivity and Trade-offs"
_ICLR.cc/2026/Conference — ICLR 2026 Conference Withdrawn Submission_

### Official Review · Reviewer_BVA6 · 2025-10-31

**Soundness:** 2
**Presentation:** 2
**Contribution:** 2
**Rating:** 2
**Confidence:** 4

**Summary:**

The authors claim to demonstrate that, for reinforcement learning (RL) policy classes exhibiting strong correlations, quantum variational circuits provide an exponential advantage in representation compared to classical networks. In this study, the authors introduce Quantum Computing Multi-Agent Proximal Policy Optimization (QC-MAPPO), where, unlike standard MAPPO, the multi-layer perceptron (MLP) is replaced with a hybrid architecture consisting of a classical MLP encoder and a parameterized quantum variational circuit. QC-MAPPO is benchmarked against classical methods such as MAPPO, Attention-MAPPO, G2Anet, and QMIX on the StarCraft Multi-Agent Challenge tasks, showing statistically significant improvements in sample efficiency and final performance on tasks requiring tight coordination among agents.

**Strengths:**

The paper tackles the highly relevant topic of multi-agent RL and is generally well-written, with its content presented in a clear and accessible manner.

**Weaknesses:**

However, the major concern with the paper and the experiments is the difference in policy learning setups between QC-MAPPO and MAPPO, which complicates the fairness of their comparison. Specifically, MAPPO trains independent policy networks for each agent, whereas QC-MAPPO connects the policies of all agents through the same quantum circuit. While this shared architecture allows QC-MAPPO to leverage inter-agent correlations more effectively, it inherently limits the flexibility of the learned policies to environments where the number of agents is fixed. This raises a conceptual question: can QC-MAPPO truly be classified as a multi-agent method, or is it better described as a centralized single policy that processes pre-processed inputs derived from individual agent observations?

Another significant issue is that the experiments appear incomplete. For example, Figure 3 shows that none of the evaluated RL methods have converged after 2 million training timesteps. Why did the authors choose to halt training at this point? Without continuing training until convergence, it remains unclear which method truly produces the best policies. It is possible that QC-MAPPO converges shortly after 2M timesteps, but methods like QMIX continue improving their policies and eventually surpass QC-MAPPO after 3M timesteps. The authors should provide further clarification and perhaps extend the experiments to address this critical limitation.

Minor Issues:
Page 1: There is a broken citation, “(Tan, 1993; ?).” This should be corrected.
Figures throughout the paper are often screenshots and difficult to read due to low resolution and poor image quality. High-quality vector graphics or improved resolutions are strongly recommended.
Page 5, Figure 3: The figure does not include shaded regions to represent confidence intervals, which would add clarity and statistical rigor to the visualized results.

**Questions:**

Why did the authors choose to halt training at 2M trainings steps?

---

### Official Review · Reviewer_qagz · 2025-10-31

**Soundness:** 3
**Presentation:** 3
**Contribution:** 2
**Rating:** 2
**Confidence:** 4

**Summary:**

This paper studies the design of a so-called quantum-classical multi-agent reinforcement learning algorithm with the key contribution centered around proving when quantum can provide a meaningful benefit in terms of sample efficiency. The theory is complemented with an experiment on a meaningful Starcraft environment.

**Strengths:**

+ The theoretical contribution is valuable and it is necessary to have such results to understand the value of quantum.
+ The noise robustness result is useful.
+ There is a need to study such quantum architectures to understand their potential.

**Weaknesses:**

- The architecture presented seems like any other quantum RL framework. I do not have clarity on why this is being called quantum-classical. Any existing quantum RL architecture does have a component implemented classically so this is not unique here.
- The discussion on resources is appreciated, but it seems to point to a major limitation with no solution in sight.
- The overall architecture is really a well-known, well-studied architecture, the use of quantum does not seem to be strongly motivated from a technical perspective.
- The experiments are very very limited, and it is difficult to make any substantial conclusions out of these experiments.
- It is not clear why MAPPO is used in the first place.

**Questions:**

1) How can you justify the small gains in sample efficiency compared to the complexity and/or cost of deploying quantum hardware?

2) Can you evaluate your system in more extensive experiments and across different environments beyond this simple Starcraft use case?

3) Can you compare with other quantum baselines and not just classical ones?

4) Can you explain how the theoretical result on noise was leveraged or used? It seems like a thought disconnected from the rest of the paper and results. You discuss it in 6.3, but that is very high level and rather trivial.

5) How many agents did you use in the simulations?

6) Why would this approach be adopted if it comes with a significant hardware cost and limited gains?

7) How did you decide on the quantum circuit you designed?

---

### Official Review · Reviewer_3Ypz · 2025-10-31

**Soundness:** 2
**Presentation:** 2
**Contribution:** 3
**Rating:** 0
**Confidence:** 4

**Summary:**

A hybrid quantum neural network based decentralized multi-agent reinforcement learning algorithm is introduced that builds on MAPPO. The method is evaluated against a number of classical MARL baselines. Lastly, the method is tested for robustness against simulated quantum hardware noise and shows graceful degradation.

**Strengths:**

The problem formulation and proposed network architecture are explained well. The paper manuscript overall is well written. A diverse number of baselines are chosen for comparison. Results  for the proposed method appear to be a significant improvement. Wall-Clock time is reported with honesty which is greatly appreciated.

**Weaknesses:**

Figures are low resolution and contain overlapping text. Overall presentation is lacking. The explanation of the proposed algorithm is very limited and a clear

Theorem 1 is missing a proper proof. Proposition 1 is stated without context. Both of them are stated rather informally and should be formalized.

Evaluation seems to be limited: only two different settings are evaluated. The results for just one of them is shown as a win rate over number of training samples. A table or boxplot summarizing all the results would improve the paper. Moreover, the baselines should include quantum network based methods. An ablation study that analyzes "whether performance gains stem from superior representation or from the implicit regularization of quantum circuits." is announced in the Introduction but nowhere to be found.

A related work section is missing altogether. Especially, a discussion of other quantum based approaches to multi-agent RL is needed.

**Questions:**

- "This advantage stems from the vastness of the Hilbert space." - Please elaborate on that statement!
- Proposition 1: What is a "global depolarizing channel" in this context?
- Figure 3: "Shaded regions represent 95% confidence intervals over 10 seeds." - There appear to be no shaded regions, are the results that similar for all the seeds?

---

### Official Review · Reviewer_jb8D · 2025-10-31

**Soundness:** 1
**Presentation:** 1
**Contribution:** 1
**Rating:** 0
**Confidence:** 4

**Summary:**

This work formulates a cooperative MARL algorithm with an information-theoretic metric under the framework of quantum machine learning, and describes an algorithm which explicitly models the correlations between agent policies.

**Strengths:**

1. The connection to quantum machine learning is interesting, and the formulation of the total correlation metric with respect to a q-qubit variational circuit seems well-motivated.
2. The discussion on qubit scaling and computational analysis is enlightening.

**Weaknesses:**

1. Please fix the broken reference in the introduction.

2. Figures

a. Text clarity: Figures 1 and 2 are unclear and the text is poorly compressed; please note that it is standard to render images in PDF format for conference submissions.

b. Image clarity: the confidence intervals in Figure 3 (left) are not distinguishable or not present.

c. Units: note that in Figure 3 (right), the units of the y-axis need to be stated in either the y-axis label or the ticks, but not both. Style guidelines are available in [1].


3. Lack of benchmarks: 3m and 8m_vs_9m are not sufficient to demonstrate superior performance on SMAC. Ideally, all or most of the SMAC maps should be tested, along with additional benchmarks. Some suggestions could include Smacv2[2], Mamujoco[3], MPE[4], Overcooked[5].

4. Baselines

a. The choice of baselines must be well-motivated. Why have an attention mechanism, a graph network, and a value decomposition algorithm been chosen as baselines? Clear reasoning must be provided for these choices beyond the fact that they include different methods of representing correlations between cooperative policies.

b. The baselines provided are not sufficient. Why were more cooperative algorithms that use mutual information metrics not compared against? [6]

c. The hyperparameters used for each experiment and each baseline are not provided.

8. >However, most interesting coordination problems require policies that do not factorize.

The authors must qualify this statement. What is meant by 'interesting coordination problems'? Do they mean difficult coordination problems that are hard to scale? What do the mean by 'policies that do not factorize'? Do they mean policies that are explicitly entangled, or simply policies that exhibit correlation? If it is the latter, it should be noted that current methods already have this property.

9. Missing background information: what do the authors mean by 'amplitude encoding'? Does this mean representing the concatenated latent vector of the agents' with normalized logits? Additionally, the mechanism of the quantum variational circuit is not clearly discussed in this work. Is is estimating the total correlation mentioned in Definition 1? While the references may contain important background information, this work must be self-contained to the extent that readers unfamiliar with [7] should be able to understand the algorithm presented.

10. Theorem 1 is not proved in this work, and would be better marked as a Definition or Lemma.

11. While the connection to quantum machine learning is interesting, its relevance to this work seems overstated. Information-theoretic measures for MARL have been used before and it must be explained clearly why this particular formulation of the total correlation is necessary, especially since the central mechanism of the algorithm is not sufficiently explained.

[1] https://iclr.cc/Conferences/2026/AuthorGuide

[2] Ellis, Benjamin, et al. "Smacv2: An improved benchmark for cooperative multi-agent reinforcement learning." Advances in Neural Information Processing Systems 36 (2023): 37567-37593.

[3] Peng, Bei, et al. "Facmac: Factored multi-agent centralised policy gradients." Advances in Neural Information Processing Systems 34 (2021): 12208-12221.

[4] Bettini, Matteo, Amanda Prorok, and Vincent Moens. "Benchmarl: Benchmarking multi-agent reinforcement learning." Journal of Machine Learning Research 25.217 (2024): 1-10.

[5] Carroll, Micah, et al. "On the utility of learning about humans for human-ai coordination." Advances in neural information processing systems 32 (2019).

[6] Jaques, Natasha, et al. "Social influence as intrinsic motivation for multi-agent deep reinforcement learning." International conference on machine learning. PMLR, 2019.

[7] Mitarai, Kosuke, et al. "Quantum circuit learning." Physical Review A 98.3 (2018): 032309.

**Questions:**

1. Why was this work presented under the framework of quantum machine learning? If the aim was to discuss the viability of a cooperative MARL algorithm within a quantum computing setting, it was not sufficiently motivated or demonstrated. Why was no discussion given to other deep learning algorithms intended specifically for the qubit setting?

---

### Note · Authors · 2025-11-12

I have read and agree with the venue's withdrawal policy on behalf of myself and my co-authors.